# Ten simple rules for making a vocabulary FAIR

**Simon J. D. Cox**[1]*, **Alejandra N. Gonzalez-Beltran**[2], **Barbara Magagna**[3], **Maria-Cristina Marinescu**[4]

**1** CSIRO Land and Water, Melbourne, Australia, **2** Science and Technology Facilities Council, Didcot, United Kingdom, **3** Environment Agency Austria, Wien, Austria, **4** Barcelona Supercomputing Center (BSC-CNS), Barcelona, Spain

* simon.cox@csiro.au

## Abstract

We present ten simple rules that support converting a legacy vocabulary—a list of terms available in a print-based glossary or in a table not accessible using web standards—into a FAIR vocabulary. Various pathways may be followed to publish the FAIR vocabulary, but we emphasise particularly the goal of providing a globally unique resolvable identifier for each term or concept. A standard representation of the concept should be returned when the individual web identifier is resolved, using SKOS or OWL serialised in an RDF-based representation for machine-interchange and in a web-page for human consumption. Guidelines for vocabulary and term metadata are provided, as well as development and maintenance considerations. The rules are arranged as a stepwise recipe for creating a FAIR vocabulary based on the legacy vocabulary. By following these rules you can achieve the outcome of converting a legacy vocabulary into a standalone FAIR vocabulary, which can be used for unambiguous data annotation. In turn, this increases data interoperability and enables data integration.

**Data Availability Statement:** All relevant data are within the manuscript.

**Funding:** The contribution of SJDC was supported through a CSIRO Strategic Project for engagement with CODATA. The contribution of BM was

## Author summary

We present ten simple rules that support converting a list of terms not currently accessible using web standards into a vocabulary conforming to the FAIR principles–Findable, Accessible, Interoperable and Reusable. In a FAIR vocabulary each term has its own persistent web-identifier, and its definition can be downloaded in both human- and standard machine-readable formats. The goal is to enable terminology to be unambiguously cited within technical datasets, in both the dataset description, or individual fields within the data, so that data can be discovered and integrated. The rules consider arrangements for governance of a terminology alongside the technical aspects related to conversion of (typically) print-based forms to standards-based knowledge representations. The rules are presented in the sequence in which they should be considered in a conversion process.

## Introduction

Environmental sustainability, global pandemics and other natural disasters are some of the challenges we are facing in the 21st century. Addressing these challenges involves analysing vast amounts of data from different sources, which is more effective when these sources are

supported through - eLTERplus, a project funded from the INFRAIA-01-2018-2019 programme of European Union's Horizon 2020 research and innovation programme under grant agreement No 871128 - OBARIS, an FFG funded project (No 887389) The funders had no role in study design, data collection and analysis, decision to publish, or preparation of the manuscript.

**Competing interests:** The authors have declared that no competing interests exist.

aggregated to find evidence-based solutions. Understanding the data, identifying the terminology used in each dataset and how the terminology in different datasets relates is a prerequisite to enable data integration.

Shared terminology is key to accurate communication and an enabler for data integration. Many organizations and disciplines have a tradition of curating lists of terms to serve various roles, particularly in metadata, column headings, and for some values in datasets. These are often called *code-lists* or *glossaries*, and if there is a process to manage them, *'controlled-vocabularies'*. Vocabularies may also be structured as hierarchies, thesauri, taxonomies, through to axiomatized ontologies [1]. Other sets of terms and codes that are used in data include units of measure, lists of materials, taxa, substances, and reference systems like geologic and dynastic time-scales (which are composed of ordered named intervals).

These vocabularies were typically managed as lists or tables within text-based resources (books and manuals), or sometimes as authority-tables in databases or in spreadsheets, for use within very specific communities and applications. We refer to these as "legacy vocabularies". However, integration of datasets, both within and across applications, requires that the terminology used in them is interoperable, so that users in the target communities (a) share an understanding of the meaning of terms, and (b) use the same conventions for representing the terms within datasets.

Standard knowledge representation languages make a vocabulary not only useful for humans, but also for machines. A number of guidelines are available for creating and publishing new vocabularies (e.g. [2,3]). Nevertheless, the legacy vocabularies represent the accumulated consensus of important disciplines and communities. Hence, making those vocabularies FAIR—or Findable, Accessible, Interoperable and Reusable [4,5]—is a high-value activity that can preserve the embedded domain intuition and knowledge. While controlled-vocabularies were often defined and used within small communities or organizations, FAIR vocabularies can be used in the context of much larger interconnected data and communities, and be actionable by machines.

Our approach to making a vocabulary FAIR is to use Web technology as outlined in the rules below. We focus on the publication of the vocabulary as 'Linked Data' which means (i) on the web, with an individual persistent resolvable unique web identifier (web link) per term (i.e. a HTTP (Hypertext Transfer Protocol) IRI (Internationalized Resource Identifier)) (ii) when a term IRI is requested, a machine-readable representation of the term using Semantic Web standards is obtained (see Table 1 for a summary of how we assess if a vocabulary is FAIR, and Box 1 for some basic definitions relating to Semantic Web standards https://www.w3.org/standards/semanticweb/data).

**Table 1. Summary of FAIR principles applied to a vocabulary.**

| | |
|---|---|
| **F** | • Each vocabulary is denoted by a persistent unique web identifier<br>• Each term is denoted by a persistent unique web identifier<br>• It is possible to search for a term or vocabulary and get a web identifier for it<br>• The vocabulary is available from at least one repository recognised by the community |
| **A** | • When the vocabulary or term identifier is de-referenced, a machine- or human-readable representation is returned, as requested |
| **I** | • At least one representation conforms to a community standard for vocabularies<br>The vocabulary includes mapping relations to other vocabularies |
| **R** | • The license for use of the vocabulary is clear and accessible<br>• Enough metadata at vocabulary and term-level is provided, including provenance and maintenance information<br>• The definitions are sufficient for a user to understand what each term means |

### Box 1. Some basic Semantic Web definitions

The Resource Description Framework (**RDF**) is the core data model of the Semantic Web. RDF-Schema (**RDFS**) is an extension of RDF and is used for representing simple RDF vocabularies on the Web. Based on RDF, the Web Ontology Language (**OWL**) is a computational logic-based language for ontologies. The Simple Knowledge Organization System (**SKOS**) is a simple OWL ontology to represent Knowledge Organization Systems (**KOS**) such as thesauri, term lists and controlled vocabularies.

To make legacy vocabularies FAIR, processes and practices are required for transitioning and adapting vocabularies from traditional forms rooted in print technologies to more broadly accessible modes that are available openly on-demand, as web resources. These have been demonstrated in many projects and services (e.g. [6]). Our goal here is to distill guidelines for taking an existing list of terms and converting it to a web-accessible, FAIR vocabulary, and present the guidelines as 'ten simple rules'.

In this paper we focus on one specific scenario, where:

1. there is a community requirement to use agreed terms in data or metadata

2. a suitable vocabulary (list of terms or codes with definitions) is available, hereafter called the *legacy vocabulary*; it was created by an organisation, person or group of people that we refer as the 'content custodian', who may also be maintaining and revising it moving forward

3. the legacy vocabulary is in the form of a print document, a digital document, or in a semi-structured form such as a spreadsheet, comma-separated value file (CSV), database table, or XML document, and is not arranged and published in a FAIR way that allows references to the terms to be resolved to learn what they mean, using standard web technology

4. no other vocabulary that is suitable for the application and acceptable to the community is published in a FAIR way either.

The Ten Simple Rules below describe how to convert that legacy vocabulary into a form that can be understood and linked on the Web, using existing, widely used practices, and also compatible with, and thus potentially able to be integrated with, related FAIR vocabularies. Some of the rules refer explicitly to the main FAIR principles, while others are basic vocabulary prerequisites. This scenario is narrow, but common. The resulting representation may not be axiomatized enough to support automated reasoning and logic operations, but publication in a form that allows specific web references is a significant improvement over the legacy forms.

We provide extensive supplementary material online at https://fairvocabularies.github.io/examples/ in the form of detailed examples taken from real vocabularies that illustrate the rules. It is strongly recommended to consult these examples in order to more fully understand details of our Ten Simple Rules.

This paper is complementary to Ten Simple Rules about vocabulary development [7] and vocabulary selection [8], and the best practices and recommendations for implementing FAIR vocabularies that primarily apply to new vocabularies rather than legacy ones that need conversion into FAIR [2,3]. The rules are arranged as a *stepwise recipe* for creating a FAIR vocabulary based on the legacy vocabulary. A partial alignment to the best practice recommendations is provided after the rules.

## Rules

### Rule 1. Determine the governance arrangements and custodian of the legacy vocabulary

Identify the *content custodian*, which is the agent (i.e. organization or person/people) that was responsible for creating or selecting the list of terms in the legacy vocabulary. They will have expertise in the subject-matter. They may be an individual, a formal or informal committee or working group, or an official organization, such as a government agency, or learned society, and will usually be managing the vocabulary on behalf of a specified community, discipline, organization, and/or jurisdiction.

When you have identified the content custodian, it is recommended that you advise them of your plan to repurpose the legacy vocabulary as a FAIR vocabulary, to get their acknowledgement of your initiative. Enrol them in the repurposing process if possible. Find out their planned revision schedule for the legacy vocabulary, so that you can allow for this in your FAIR vocabulary maintenance plan (Rule 10).

### Rule 2. Verify that the legacy-vocabulary license allows repurposing, and agree on the license for the FAIR vocabulary

Verify that the copyright-holder grants permission for the list of terms to be re-published as 'Linked Data' (noting that the copyright-holder is often different to the maintainer or content custodian—see Rule 1).

If the source carries a Creative Commons license, then the No Derivatives (ND) options (CC BY-**ND**, CC BY-NC-**ND)** are *not* ok, since you are developing a 'derivative product'.

The other CC licenses (CC0, and CC BY, CC BY-SA, CC BY-NC, CC BY-NC-SA) are suitable, provided you are also able to meet any BY (attribution), SA (share-alike) and NC (non-commercial) constraints.

If the original content uses another type of license, you must analyse it to understand if you are able to produce a derivative product, and what are the conditions for derivation. It may be necessary to contact the copyright-holder directly in order to explain what is planned and get permission.

Agree on the license for the FAIR vocabulary, preferably an open license for users (e.g. CC0 or CC-BY).

### Rule 3. Check term and definition completeness and consistency in the legacy vocabulary

Ensure there is at least (i) a unique label and (ii) a description or textual definition for each term in the list. These are the minimum requirements for a useful vocabulary, and the minimum required information for encoding the FAIR vocabulary (Rule 6). Verify that the definitions are unambiguous, and ideally that they are distinct. If definitions overlap, or are missing or ambiguous, consult the custodian of the legacy vocabulary and ensure that the representation follows the reality of the domain (Rule 1), else identify or recruit an expert group to revise, review or provide definitions and sources. Ideally this should be composed of more than one person to allow a quality control cycle. As a last resort check with a public source for definitions such as Wikipedia, DBpedia, or Wikidata.

The legacy vocabulary may also contain synonyms, intra-vocabulary relationships such as a broader/narrower hierarchy, specified subsets, and cross-vocabulary mappings. Guidelines to encode all of these elements in a FAIR vocabulary are given in Rule 6.

### Rule 4. Establish a traceable maintenance-environment for the FAIR vocabulary content

It is common to store the reference version of the FAIR vocabulary in a single file, using one of the standard RDF serializations (e.g. Turtle, RDF-XML, JSON-LD). It is strongly recommended to maintain this in a system that allows any changes made in the vocabulary to be easily traced. Thus, we recommend use of a version control system (e.g. BitBucket, GitHub, GitLab). Public access should be allowed, unless the content owner has good reason not to. An issue tracker or ticket system should be used to capture term requests or other proposals by members of the community, and to record the justification for individual changes made by the content custodian.

Note that an issue tracker is built into GitHub and GitLab; JIRA or Trac are popular stand-alone options.

More details on reflecting changes and revisions to the vocabulary content out to the published FAIR vocabulary are discussed in Rule 10.

### Rule 5. Assign a unique and persistent identifier to (a) the vocabulary and (b) each term in the vocabulary

Choose a domain name for persistent identifier IRIs for the terms and other vocabulary items (e.g. collections of terms). Those IRIs must resolve to appropriate representations on the web over the lifetime of any datasets that will make use of them, so it should be planned to manage this domain over a 10+ year time period. Since this is longer than many organization names and most organizational structures, domain names based on organizations are generally *not* suitable, except if they are of organizations specifically created for the purpose of managing vocabularies. Consider existing open solutions for persistent identifiers such as https://w3id. org or http://purl.org as an alternative to managing your own HTTP server.

Choose and document the pattern for individual IRIs that identify terms in the vocabulary [9,10]. A common pattern is:

$$IRI = [http : //|https : //] + \{domain\} + \{vocab\} + \{term{-}id\}$$

where {domain} is the long-lasting host for the FAIR vocabulary, {vocab} is a path composed of a sequence of tokens separated by slash characters ('/'), and {term-id} denotes the individual term, and must be unique in the context of the vocabulary. Some complete IRIs for terms that demonstrate this pattern are

http://anzsoil.org/def/au/asls/landform/modal-slope
http://resource.geosciml.org/classifier/ics/ischart/Cambrian
http://vocabs.lter-europe.net/EnvThes/21279
http://purl.obolibrary.org/obo/ENVO_00000081
http://qudt.org/vocab/unit/DEG_C
http://vocab.nerc.ac.uk/collection/P06/current/UPAA/

The {term-id} may be an opaque code (e.g.numeric), or it may be based on the term or primary label for each term, or some other rule. For vocabularies with up to a few hundred terms where the meanings do not change over time, use of a label as the basis for a {term-id} may be manageable, and this can be a useful mnemonic for developers and maintainers. However, it is important to consider the stability of the current label, and have a strategy for managing the IRI if a different label becomes preferred for the same concept. For large vocabularies, or when labels may change over time, label-based patterns are difficult to sustain for the {term-id}, and numeric or opaque identifiers are more common [9].

It is recommended not to embed version information in the path or identifier, as this creates challenges if the same concept persists over multiple versions or releases.

It is recommended to avoid long paths. Hierarchical relationships should not be implied by the IRI path, but rather should be recorded explicitly within the representation of the term (see Rule 6).

It is recommended to use slash ('/') IRIs for large vocabularies, rather than hash ('#') IRIs. When a # IRI is requested the entire vocabulary will be returned instead of just a single term. This may be acceptable for a small vocabulary, but is undesirable for large vocabularies [10].

In Rule 9 we outline how the IRIs should be made resolvable, thus making the vocabulary, and its terms, accessible.

For more examples, see the online supplementary material https://fairvocabularies.github.io/examples/

## Rule 6. Create machine readable representations of the vocabulary terms

Convert the vocabulary to semantic standards, using either the Simple Knowledge Organisation System (SKOS) [11–13] or the Web Ontology Language (OWL) [14,15], together with elements from other standard vocabularies and ontologies where appropriate (e.g. Dublin Core [16,17]).

The table below details various technical steps and patterns for use of either SKOS or OWL to represent a vocabulary in RDF. There are a number of considerations in making a choice of one or the other of these pathways [18]:

- SKOS was designed for sets of definitions optionally arranged in a hierarchy, so nicely fits the primary scenario under consideration here: i.e. conversion of a legacy vocabulary to an RDF-based form using a semi-formal representation. SKOS includes a number of features designed to make the conversion straightforward, including synonyms, codes, subsets, and broader/narrower relationships. However, there are limitations in its logical completeness that are considered weaknesses in some applications;

- OWL supports axiomatization (based on description logics) for representing formal ontologies, and was designed for a much wider range of applications than the primary scenario. However, the design choices using OWL are complex, and describing them is well beyond the scope of this paper. Nevertheless, a basic OWL pattern is outlined below, with the namespaces limited to core vocabularies. This option most closely parallels SKOS, and is thus suitable for the primary use-case covered by this paper.

Other sources provide details on SKOS and OWL, their particular strengths, and how they can be used together (e.g. [13,19]). We include the OWL option here because a rich OWL representation is a potential future goal for a FAIR vocabulary, so a minimal version is a useful starting point. However, the choice of representation is not critical in this phase of vocabulary formalization.The most important feature is that a unique IRI is used to denote each distinct term (see Rule 5), so that these IRIs can be used in data or metadata. The representation of each term might be changed or supplemented later while retaining the same IRI, and alternative representations or descriptions can be provided to suit each application, as long as they describe the same underlying concept (see Rule 9).

Table 2 illustrates basic steps to follow to create a FAIR vocabulary relying on SKOS or OWL (for the namespace prefixes see Box 2). The SKOS terminology is standard. For an OWL representation we suggest some common elements, and for more expressive ontologies we recommend investigating OWL and the conventions of the community you want to target.

In Box 3. we show an example of both representations side-by-side for the same term (serialized in Turtle[21]):

**Table 2. Steps in the creation of machine-readable definitions.**

| Step | SKOS | OWL (basic) |
|---|---|---|
| **Identify terms** | Encode each vocabulary term as a skos:Concept, assigning an identifier as discussed in Rule 5 | Encode each vocabulary term as an owl:Class, assigning an identifier as discussed in Rule 5 |
| **Encode term labels and synonyms** | Encode the term name as the skos:prefLabel and synonyms and abbreviations in skos:altLabel.<br>Language tags [20] can be used for multilingual vocabularies. | Encode the term name as rdfs:label. For synonyms you can use the SKOS elements skos:prefLabel and skos:altLabel.<br>Note that some communities have their own terminology for labels and annotations (e.g. the OBO Foundry relies on the Information Artifact Ontology).<br>Language tags [20] can be used for multilingual terms. |
| **Add textual definitions** | Encode the textual definition as skos:definition | Encode the textual definition as rdfs:comment |
| **Add codes and symbols** | Add any code or symbol as skos:notation (this applies if a formal code or symbol for the term is available, in addition to the name used for the skos:prefLabel) | Add any code or symbol as additional labels (rdfs:label) |
| **Add notes or comments for clarifications** | Comments can be encoded using skos:note. Clarifications on usage can be recorded using skos:scopeNote | Comments and clarifications can be encoded in the rdfs:comment |
| **Add per-term metadata, if available** | Individual terms may be annotated using standard elements such as dcterms:creator, dcterms:created, dcterms:identifier, dcterms:modified, dcterms:source, dcterms:replaces, rdfs:seeAlso.<br>Per-term annotations are needed when they differ from values associated with the vocabulary as-a-whole (see Rule 7). | The same metadata elements can be used to annotate terms in the OWL encoding as well as owl:versionInfo, rdfs:comment, rdfs:isDefinedBy. Alternatively, adopt a specific solution for describing term metadata such as the OBO Metadata Ontology (http://www.obofoundry.org/ontology/omo.html) |
| **Define the hierarchy of terms** | If hierarchical relationships between terms are provided in the source document, encode these using skos:broader and skos:narrower. A narrower concept or subclass may be related to more than one broader concept or parent class, so each term may appear in more than one place in a hierarchy. | If hierarchical (is-a-kind-of) relationships between terms are provided in the source document, encode these using rdfs:subClassOf. A more specific concept or subclass may be related to more than one broader concept or parent class, so each term may appear in more than one place in a hierarchy. N.B. OWL sub-class relationships have more precise semantics than SKOS narrower/broader relations. |
| **Encode relationships between terms** | If other relationships (non-hierarchical) between terms *within* the vocabulary are provided in the source document, they may be encoded using skos:related.<br>Relationships to terms in *other* vocabularies (mappings) may be encoded using skos:broadMatch, skos:closeMatch, skos:exactMatch, skos:narrowMatch, skos:relatedMatch | dcterms:relation may be used to indicate related resources. However, it is usually better to use OWL object properties with the specific required semantics. It is recommended to re-use existing elements (e.g. relations ontology http://www.obofoundry.org/ontology/ro.html), or create your own if no existing one fulfils your requirement.<br>owl:equivalentClass may be used for mappings. |
| **Define subsets** | If subsets or other groupings of terms are present in the source, encode each as a skos:Collection.<br>Collections may be nested.<br>Concepts may be members of more than one collection. | If subsets or other groupings of terms are present in the source, encode each as a class whose members are sub-classes |
| **Define the whole vocabulary** | The complete vocabulary should be encoded as a skos:ConceptScheme. Every skos:Concept should have a skos:inScheme relationship to the scheme, else the top terms in broader/narrower chains should have a skos:topConceptOf relationship to the concept scheme. | The complete vocabulary should be encoded as an owl:Ontology. Every member term that is not in the same namespace as the ontology should have a rdfs:isDefinedBy relationship to the ontology |

## Box 2. Namespace prefixes mentioned in Rule 6

dcterms: http://purl.org/dc/terms/

owl: http://www.w3.org/2002/07/owl#

rdf: http://www.w3.org/1999/02/22-rdf-syntax-ns#

rdfs: http://www.w3.org/2000/01/rdf-schema#

skos: http://www.w3.org/2004/02/skos/core#

### Box 3. SKOS and OWL representations of the same term

```
ex:element-80 a skos:Concept;
   skos:prefLabel "mercury"@en;
   skos:prefLabel "mercurio"@es;
   skos:altLabel "quicksilver"@en;
   skos:notation "Hg";
   skos:definition "A heavy, silvery d-block element, mercury is the only
metallic element that is liquid at standard conditions for temperature and
pressure.";
   dcterms:identifier "7439-97-6";
   dcterms:source <https://en.wikipedia.org/wiki/Mercury_(element)>;
   skos:broader ex:group-12, ex:period-6;
   skos:exactMatch <http://purl.obolibrary.org/obo/CHEBI_16170>;
   skos:inScheme ex:periodicTable;
.
```

```
ex:element-80 a owl:Class;
   rdfs:label "mercury"@en;
   rdfs:label "mercurio"@es;
   skos:altLabel "quicksilver"@en;
   rdfs:label "Hg";
   rdfs:comment "A heavy, silvery d-block element, mercury is the only
metallic element that is liquid at standard conditions for temperature and
pressure.";
   dcterms:identifier "7439-97-6";
   dcterms:source <https://en.wikipedia.org/wiki/Mercury_(element)>;
   rdfs:subClassOf ex:group-12, ex:period-6;
   owl:equivalentClass <http://purl.obolibrary.org/obo/CHEBI_16170>;
   rdfs:isDefinedBy ex:periodicTable;
.
```

Different approaches will be required for the conversion, depending on the form of the source material.

- Where the original vocabulary is only available as a printed document, scanning, or even rekeying the essential information may be the only practical route; if available as a digital text document, you may be able to copy and paste the information

- Where the legacy vocabulary is tabulated, either fully or in part, it may be possible to identify a pattern or template from the elements of your vocabulary which will allow you to (fully or partly) automate the creation of the FAIR vocabulary. Tools such as *SKOS-Play*! or *sheet2rdf* and *OpenRefine* can convert spreadsheets to RDF. Links to these, and to tools to convert many other formats to RDF are available at https://www.w3.org/wiki/ConverterToRdf.

- *qSKOS* (https://qskos.poolparty.biz/) is a useful structure- and quality-checker for SKOS vocabularies, and SKOSify (https://skosify.readthedocs.io/en/latest/) automates some conversion and cleaning operations.

- *Ontorat* [22] and ROBOT [23] can be used for generating terms, annotations and axioms of an OWL vocabulary based on ontology design patterns or templates; in addition, ROBOT has other functionality to automate ontology development workflows.

Either way, it is recommended to use an RDF/OWL or SKOS IDE (Integrated Development Environment) such as *TopBraid*, *Protégé*, *VocBench*, or *PoolParty* for data entry, or for tidying up after an automated phase, and for consistency checking.

The FAIR vocabulary should represent the legacy vocabulary as closely as possible, so it is *not* recommended to change the vocabulary content or structure during encoding, even if there appear to be errors or potential improvements. The initial FAIR representation can serve as a baseline for future revisions, while clearly anchored to an archival source. Changes to the content of the legacy and FAIR vocabularies remain the prerogative of the content custodian identified in Rule 1, and the maintenance process described in Rule 10.

### Rule 7. Add vocabulary metadata

Add metadata for the vocabulary, by adding metadata elements to the skos:ConceptScheme or owl:Ontology that represent the vocabulary-as-a-whole.

The description of the vocabulary must include at least:

- provenance and ownership information (citation of or links to the source, pointers to the organization or community responsible for the content),

- lifecycle information (creation and update dates, vocabulary status, pointers to the people responsible for the conversion and encoding, version information)

- Vocabulary license, as agreed in Rule 2

Different communities rely on metadata elements as defined by different vocabularies such as Data Catalog Vocabulary (DCAT [24]), Linked Open Vocabularies (LOV [25]), Ontology Metadata Vocabulary (OMV [26]), or the Metadata for Ontology Description and Publication Ontology (MOD [27]). OWL includes some built-in annotation properties that are applicable to OWL ontologies (e..g owl:priorVersion, owl:backwardsCompatibleWith, owl:incompatible-With). The choice of which metadata vocabulary and details about mandatory requirements should be prescribed in policies of the vocabulary repository (Rule 8), as well as documented in the metadata for the vocabulary with full text or a link to a policies document.

## Rule 8. Register the vocabulary

Load or register the encoded content in a vocabulary service or semantic repository, such as Research Vocabularies Australia (RVA) (https://vocabs.ardc.edu.au/) (for SKOS vocabularies), Linked Open Vocabularies (LOV) (https://lov.linkeddata.es/dataset/lov/ [28]) (for OWL ontologies and SKOS vocabularies), the ESIP Community Ontology Repository (https://cor.esipfed.org/) or BioPortal (https://bioportal.org) and its derivatives such as Agroportal (http://aims.fao.org/agroportal) and Ecoportal (http://ecoportal.lifewatchitaly.eu/ontologies) (for OWL ontologies and SKOS vocabularies). If you expect to be maintaining many vocabularies you might establish your own service using one of the software stacks available.

You should also deposit release snapshots of the vocabulary in a repository such as Zenodo (https://zenodo.org) or Dryad (https://datadryad.org/stash), or in an institutional data repository available to you. This step will assign a DOI to the vocabulary and will ensure that the vocabulary is indexed in more general search engines. See Rule 4 for recommendations of using a version control system, and consider that there are automated ways to store Github releases in Zenodo (with associated DOI). You may also consider registering the FAIR vocabulary as a 'standard' in FAIRsharing (https://fairsharing.org/).

Finally, the community for whom the vocabulary is provided (identified in Rule 1) is likely to maintain a listing of community resources, which is often the first place that members of the community would look. Such venues would be a good target for linking to the vocabulary.

## Rule 9. Make the vocabulary accessible for humans and machines

The web identifiers used in the vocabulary should resolve to specific digital objects. Thus, the HTTP server for the vocabulary domain (identified in Rule 5) must be configured so that any request for an IRI denoting a term gets a representation of the individual term from the service that hosts the vocabulary. Use standard HTTP content negotiation to provide access to different representations (using Accept: and Accept-profile: headers [29]). The representation should be a web page (if HTML is requested) or a serialized skos:Concept or owl:Class (if RDF is requested). The IRI for the vocabulary-as-a-whole should get a suitable 'Landing Page' (if HTML is requested) or a representation of the skos:ConceptScheme or owl:Ontology (if RDF is requested). The HTML representation can be generated automatically with existing tools (e.g. [30]). The representation should include metadata and attribution information. (Note that inbuilt metadata means that there is no advantage to licensing the FAIR vocabulary with CC-BY compared with CC0 (see Rule 2).)

SPARQL [31,32] is the standard RDF query interface, so a SPARQL endpoint may be provided to support flexible queries and interactions. A link to the SPARQL endpoint should be

provided on the HTML landing pages. The public endpoint should not allow SPARQL Update operations [33]. The hosting service may provide other vocabulary Application Programming Interfaces (e.g. RVA provides SISSvoc [34]). These should be clearly advertised to the user-community.

## Rule 10. Implement a process for publishing revisions of the FAIR vocabulary

The FAIR vocabulary should be created and maintained so that it reflects the content and updates agreed and issued by the content custodian, so it is important to obtain the maintenance schedule and versioning strategy for the vocabulary from the content custodian (Rule 1).

We recommend updating the FAIR vocabulary as soon as practical after the content custodian updates the legacy vocabulary. If the content custodian wishes to maintain the content in its original form (i.e. the legacy vocabulary), then try to arrange for alerts advising you of changes to be issued by the custodian, in order to trigger the process of update of the FAIR vocabulary. However, it may be possible to transition to an arrangement in which the FAIR vocabulary becomes the primary version or 'point of truth' for the content, in which case individual revisions should be proposed and tracked in a traceable maintenance environment (see Rule 4). However, this should only be done with the consent of the content custodian. Note that as well as improved tracking of revisions, some kinds of improvement may be supported better in the FAIR representation (see Rule 6) than on the legacy (print-based) platform, including specific relationships between terms, mappings to other vocabularies, and detailed axiomatization of definitions.

If revision of the vocabulary is by new releases of the vocabulary-as-a-whole, then status and version information will be in the vocabulary metadata (see Rule 7). If maintenance is continuous, then the per-term metadata should capture its status and version information (see Rule 6). Standard Dublin Core, SKOS and OWL properties that may be useful in versioning include:

- dcterms:created—date or date-time that the vocabulary or term was initially created

- dcterms:modified—date or date-time that the vocabulary or term was last updated

- dcterms:isReplacedBy—to point to a superseding vocabulary or term

- dcterms:replaces—to point to a prior version of a vocabulary or term

- owl:deprecated = 'true' if the vocabulary or term is no longer valid

- owl:priorVersion—to point to a previous version of a vocabulary

- owl:versionInfo—general annotations relating to versioning

- skos:changeNote—modifications to a term relative to prior versions

- skos:historyNote—past state/use/meaning of a term

Do not re-assign or remove identifiers; they are persistently associated with the term to which they were originally assigned (Rule 5). If necessary, you can deprecate or retire an identifier. However, the IRI for every retired and superseded term must remain de-referenceable, as well as for previous versions of the vocabulary, so that references to them still return a result, annotated with the status.

Terms that carry over between releases without the definition changing must retain the same IRI. If the IRI were changed, then datasets that use different versions of the same

vocabulary cannot interoperate. Consult with the content custodian to clarify the 'identity-determining' characteristics of terms, but note that changing relationships (e.g. position in a hierarchy) or the textual definition do not *necessarily* require changing the identifier (i.e. minting a new IRI) provided that the intention for the concept is still the same.

## Alignment with other guidelines

We mentioned existing guidelines that focus primarily on the development of new vocabularies. In Table 3 we align our Ten Simple Rules with the recommendations and practices from two of these [2,3] as well as with the W3C Data on the Web Best Practices [35]. The alignment is only partial, as the other work goes into more detail on some topics, while some of the concerns discussed in our rules are not addressed in the other work.

**Table 3. Alignment of the Ten Simple Rules with some other best practices.**

| | Ten Simple Rules | FAIRsFAIR [3] | Best practices—Garijo & Poveda [2] | Data on the Web Best Practice [35] |
|---|---|---|---|---|
| 1 | Determine the governance arrangements and custodian of the legacy vocabulary | BP-Rec 7—Interact with the designated community and manage user centric development<br>BP-Rec 9 -The underlying logic of semantic artefacts should be grounded on the domain it intends | | 33—Provide Feedback to the Original Publisher |
| 2 | Verify that the legacy-vocabulary license allows repurposing, and agree on the license for the FAIR vocabulary | P-Rec16—The semantic artefact should be clearly licenced for machines and humans | | 4—Provide data license information<br>34—Follow Licensing Terms |
| 3 | Check term and definition completeness and consistency in the legacy vocabulary | BP-Rec 8—Provide a structured definition for each concept<br>BP-Rec 9—The underlying logic of semantic artefacts should be accurately grounded on the domain it intends to describe | | |
| 4 | Establish a traceable maintenance-environment for the FAIR vocabulary content | BP-Rec 10—Define a set of governance policies for the semantic artefacts | | |
| 5 | Assign a unique identifier to (a) the vocabulary and (b) each term in the vocabulary | P-Rec 1—Use Globally Unique, Persistent and Resolvable Identifier for Semantic Artefacts, their content and their versions<br>BP-Rec 1—Use a unique naming convention for concept/class and relations<br>BP-Rec 2—Use an Ontology Naming Convention | 2.1—Ontology name and prefix<br>2.2—Hash versus slash URIs<br>2.5—Using permanent URIs | 9—Use persistent URIs as identifiers of datasets<br>10—Use persistent URIs as identifiers within datasets<br>27- Preserve identifiers |
| 6 | Create machine readable representations of the vocabulary terms | P-Rec 3—Use a common minimum metadata schema to describe semantic artefacts and their content<br>P-Rec 9—Semantic artefacts should be compliant with Semantic Web and Linked Data standards<br>P-Rec 11—Use a standardized description for complex logical relations<br>P-Rec 14—Use standard vocabularies to describe semantic artefacts<br>BP-Rec 3—Use defined ontology design patterns<br>BP-Rec 6—Harmonize the methodologies used to develop semantic artefacts<br>BP-Rec 8—Provide a structured definition for each concept | | 12—Use machine-readable standardized data formats<br>16—Choose the right formalization level |
| 7 | Add vocabulary metadata | P-Rec 3—Use a common minimum metadata schema to describe semantic artefacts and their content<br>P-Rec 17—Provenance should be clear for both humans and machines | 3.1 Ontology Metadata | 1—Provide metadata<br>2—Provide descriptive metadata<br>5—Provide data provenance information<br>7—Provide a version indicator<br>8—Provide version history<br>35—Cite the Original Publication |

*(Continued)*

**Table 3.** (Continued)

| | Ten Simple Rules | FAIRsFAIR [3] | Best practices—Garijo & Poveda [2] | Data on the Web Best Practice [35] |
|---|---|---|---|---|
| 8 | Register the vocabulary | P-Rec 4—Publish the Semantic Artefact and its content in a semantic repository | 4.2 Making an Ontology Findable on the Web | |
| 9 | Make the vocabulary accessible for humans and machines | P-Rec 4—Publish the Semantic Artefact and its content in a semantic repository<br>P-Rec 5—Semantic repositories should offer a common API to access Semantic Artefacts and their content in various serializations for both use/reuse and indexation by search engines | 3.2 Creating a Human-Readable Documentation<br>3.3 Ontology visualization<br>4.1 Ontology Accessibility in Multiple Interoperable Formats | 17—Provide bulk download<br>19—Use content negotiation for serving data available in multiple formats<br>20—Provide real-time access<br>23—Make data available through an API<br>24—Use Web Standards as the foundation of APIs<br>32—Provide Complementary Presentations |
| 10 | Implement a process for publishing revisions of the FAIR vocabulary | P-Rec 8—Define human and machine-readable persistency policies for metadata<br>BP-Rec 7—Interact with the designated community and manage user-centric development<br>BP-Rec 9—The underlying logic of semantic artefacts should be accurately grounded on the domain it intends to describe<br>BP-Rec 10—Define a set of governance policies for the semantic artefacts | 2.4—Ontology versioning<br>2.5—Using permanent URIs | 21—Provide data up to date<br>10—Use persistent URIs as identifiers within datasets<br>27—Preserve identifiers<br>33—Provide Feedback to the Original Publisher |

Note that our Ten Simple Rules are ordered in a natural implementation workflow for the primary scenario, i.e. the conversion of existing vocabularies. This means that some recommendations that are grouped together in other guidelines are separated here. The sequence of Ten Simple Rules is designed for a specific audience, i.e. people assisting domain specialists, neither of whom are semantics or web specialists.

## Summary and conclusion

We have presented ten simple rules that support converting a legacy vocabulary—a list of terms available in a print-based glossary or table not accessible using web standards—into a FAIR vocabulary. Various pathways may be followed to publish the FAIR vocabulary, but we emphasise particularly the goal of providing a distinct IRI for each term or concept. A standard representation of the concept should be returned when the individual IRI is de-referenced, using SKOS or OWL serialised in an RDF-based representation for machine-interchange, or in a web-page for human consumption. Guidelines for vocabulary and term metadata are provided, as well as development and maintenance considerations.

By following these rules you can achieve the outcome of converting a legacy vocabulary into a standalone FAIR vocabulary, which can be used for unambiguous data annotation. In turn, this increases data interoperability and enables data integration, which is essential for addressing global challenges such as environmental sustainability, and pandemic and natural disaster response. A set of examples illustrating the application of these rules are provided as supplementary material at https://fairvocabularies.github.io/examples/. These include environmental definitions that are needed to cover some of the data integration challenges that we referred to in the introduction.

Further steps towards broader interoperability that may be considered, but are beyond the scope of this paper, include:

- relationships to terms and definitions in other FAIR vocabularies

- patterns for re-use of terms from and subsets of existing FAIR vocabularies

- supplementation of generic SKOS/OWL encoding with domain-based elements and axiomatization (see examples in the supplementary material)

- rules for maintenance (expanding on Rules 1, 4 & 10)

These will be addressed in future guidelines.

## Acknowledgments

We thank CODATA (https://codata.org) and the DDI Alliance (https://ddialliance.org/), who organised a Workshop on Cross-domain Metadata at Schloss Dagstuhl in October 2019, where this work was initiated. The FAIR vocabulary practices activity which triggered the preparation of this guideline initially also involved Pier Luigi Buttigieg, Niklas Kolbe, and Dan Brickley.

## Author Contributions

**Conceptualization:** Simon J. D. Cox, Alejandra N. Gonzalez-Beltran, Barbara Magagna, Maria-Cristina Marinescu.

**Resources:** Simon J. D. Cox, Alejandra N. Gonzalez-Beltran.

**Writing – original draft:** Simon J. D. Cox, Alejandra N. Gonzalez-Beltran.

**Writing – review & editing:** Simon J. D. Cox, Alejandra N. Gonzalez-Beltran, Barbara Magagna, Maria-Cristina Marinescu.

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
