## [Decision Letter · Decision Letter 0]

31 Dec 2020

Dear Dr Cox,

Thank you very much for submitting your manuscript "Ten Simple Rules for making a vocabulary FAIR" for consideration at PLOS Computational Biology.

As with all papers reviewed by the journal, your manuscript was reviewed by members of the editorial board and by several independent reviewers. In light of the reviews (below this email), we would like to invite the resubmission of a significantly-revised version that takes into account the reviewers' comments.

We cannot make any decision about publication until we have seen the revised manuscript and your response to the reviewers' comments. Your revised manuscript is also likely to be sent to reviewers for further evaluation.

Sincerely,

Scott Markel, Ph.D.

Ten Simple Rules Editor

PLOS Computational Biology

Scott Markel

Ten Simple Rules Editor

PLOS Computational Biology

Reviewer's Responses to Questions

**Comments to the Authors:**

Reviewer #1: Two Documents attached:

1. pdf of manuscript with comments using Acrobat

2. pdf of document with comments and suggestions (CoxEtAl10RulesVocabularySMR.pdf)

Reviewer #2: The authors present here a succinct and comprehensive set of advise for both encoding vocabularies in Semantic Web standards and for making vocabularies fit well into the FAIR principles for Data Management. Having been responsible for publishing a number of controlled vocabularies, I find their steps here to a well laid out Best Practice. As such this is a timely paper and the authors are to be congratulated for their input and presentation. I have only one very minor comment, which is that on lines 42, 94 and 95 the render the phrase "linked data" and "linked-data", which I would more normally expect to see as "Linked Data".

Once again - thank you for the opportunity to review this well written paper.

Reviewer #3: This paper proposes a set of 10 rules to provide FAIR vocabularies. The work is interesting and worth being published as FAIR data and related initiatives are gaining momentum. In this sense, the first recommendation for improving the paper would be to align the proposed rules to the previous works as "Best Practices for Implementing FAIR Vocabularies and Ontologies on the Web" (http://ebooks.iospress.nl/volumearticle/56005 and arxiv version https://arxiv.org/abs/2003.13084) and the FAIRsFAIR D2.2 FAIR Semantics: First recommendations (https://zenodo.org/record/3707985). Authors may also be interested in the special issue https://www.mitpressjournals.org/doi/full/10.1162/dint_e_00023

Other important comments to enhance the current paper are:

Line 171: What do you mean by "Resource Description Framework (RDF) vocabularies"? the references mix the RDF and RDFS recommendations with Dublin Core. RDF is the data model for the web, RDFS is the language for vocabularies and OWL is a language for ontologies. Finally Dublin Core is a controlled vocabulary that happens to be implemented in RDFS and in OWL (depending elements or terms.) Please clarify these concepts. It would be advisable also to clarify in general than OWL is the ontology language and SKOS is an OWL ontology to represent thesaurus, not a particular language itself as in general lines from 175 to 187 might be a bit misleading, in particular lines 185-187 as the constructs that are "missed" in OWL are actually built in SKOS based on OWL and the reasoning features are defined in OWL.

Regarding the table for steps for SKOS and OWL conversion I would suggest to keep it just for SKOS as there are several points in the OWL column. In general I feel that that column could lead to conceptual errors to the readers, overall not experts in knowledge representation, as it seems to oversimplify a great amount of work about ontological engineering. While the steps for converting to a SKOS are clear and could be automated, for building an ontology there are a number of consideration that should be made and given the type of publication it would be more suitable to keep it to SKOS rather than entering in an ontological engineering work. The points about the table are:

-- "Identify terms": "or as an instance of an owl:Class if it is the most specific concept in the vocabulary "  not always, one may want to have all as classes and create individual for the last level. It depends on the use.

-- "Encode term labels and synonyms"  In general, it is a bad practice to use the owl:equivalentClass construct to define synonyms, they could be added as several labels to a given class or with other annotations but not creating N classes in the same ontology unless there is a good reason for that.

-- "Define the hierarchy of terms"  transforming broader/narrower relationships to rdfs:subClassOf would lead to semantic errors and reasoning issues. See that in a thesaurus we can have PC - narrower - mouse. If we make mouse subclass of PC is a semantic inconsistency as individuals of mouse will be classified as individuals of PC. In this cases a transformation is not that simple and if one wants to transform a thesaurus into an ontology there should be some modelling decisions to be taken.

-- minor: rdfs:comment is used in steps 3 and 5 for different purposes, the original type of annotation would be lost.

-- "Add per-item metadata": for the case of ontologies the LOV, MOD https://github.com/sifrproject/MOD-Ontology and Widoco guidelines could be included but as said, I would suggest keeping the table for SKOS.

Other comments are:

In rule 8 LOV should be added as registry as it is widely used by many communities and Ontobee.

Line 32 and 61: the term glossary seems to be used as synonym for controlled vocabularies and could be a bit inaccurate. I'd suggest using and referring to the ontology spectrum by McGuinness http://www.ksl.stanford.edu/people/dlm/papers/ontologies-come-of-age-mit-press-(with-citation).htm

Line 61 to 65: I don't see why those example "may not be initially recognised as vocabularies" if word-lists are vocabularies list of terms which are unit of measure are too.

Minor comments:

Line 73 to 76: the list of digital documents is too narrow, it might include relational databases, XML files, etc.

Line 81: ".." remove one "."

**Have all data underlying the figures and results presented in the manuscript been provided?**

Reviewer #1: Yes

Reviewer #2: Yes

Reviewer #3: None

PLOS authors have the option to publish the peer review history of their article (what does this mean?). If published, this will include your full peer review and any attached files.

Reviewer #1: **Yes: **Stephen M Richard

Reviewer #2: No

Reviewer #3: No
---

## [Decision Letter · Decision Letter 1]

8 Apr 2021

Dear Dr Cox,

Thank you very much for submitting your manuscript "Ten Simple Rules for making a vocabulary FAIR" for consideration at PLOS Computational Biology. As with all papers reviewed by the journal, your manuscript was reviewed by members of the editorial board and by several independent reviewers. The reviewers appreciated the attention to an important topic. Based on the reviews, we are likely to accept this manuscript for publication, providing that you modify the manuscript according to the review recommendations.

Sincerely,

Scott Markel, Ph.D.

Ten Simple Rules Editor

PLOS Computational Biology

Scott Markel

Ten Simple Rules Editor

PLOS Computational Biology

[LINK]

Reviewer's Responses to Questions

**Comments to the Authors:**

Reviewer #1: The revised version is excellent. Thanks for putting this together!

Reviewer #2: This revision looks excellent - I recommend for publishing

Reviewer #3: Thank you very much for your detailed comments and the effort improving the paper.

There are substantial changes in the manuscript addressing the reviewer's comments. I think the paper has improved and actually it a good starting point for converting legacy terminologies to FAIR vocabularies.

I only have some small comments:

Line 73: is there an "and" or "or" missing before "(b)"?

Lines 226 to 229: It is common to use # for small vocabularies or ontologies, it would be better to have that mentioned. I understand it is different if you are transforming a huge KOS.

Table skos and owl, point "Define the hierarchy of terms" in the owl column: "A narrower concept or subclass"  I would avoid mentioning here "narrower", maybe "A more specific concept or subclass... ". Also, I would add a clarification stating that in owl the subclass is oriented to subsets of elements which is not exactly the same as "narrower/broader" for SKOS.

Table skos and owl, point "Define the whole vocabulary" about the sentence: "Every member term should have a rdfs:isDefinedBy relationship to the ontology"  Could this be optional for when the element URI is defined in the same namespace as the ontology URI?

Regarding the options to transform vocabularies to SKOS or OWL, I find very useful OpenRefine. Not sure how easy to use is for non-experts in RDF or semantic technologies.

**Have the authors made all data and (if applicable) computational code underlying the findings in their manuscript fully available?**

Reviewer #2: Yes

PLOS authors have the option to publish the peer review history of their article (what does this mean?). If published, this will include your full peer review and any attached files.

Reviewer #1: **Yes: **Stephen M Richard

Reviewer #2: No

Reviewer #3: No

**Have all data underlying the figures and results presented in the manuscript been provided?**

Reviewer #1: Yes

Reviewer #3: Yes

Figure Files:

Data Requirements:

Reproducibility:

References:

---

## [Editor Report · Decision Letter 2]

4 May 2021

Dear Dr Cox,

We are pleased to inform you that your manuscript 'Ten Simple Rules for making a vocabulary FAIR' has been provisionally accepted for publication in PLOS Computational Biology.

Best regards,

Scott Markel, Ph.D.

Ten Simple Rules Editor

PLOS Computational Biology

Scott Markel

Ten Simple Rules Editor

PLOS Computational Biology

---

## [Editor Report · Acceptance letter]

11 Jun 2021

PCOMPBIOL-D-20-02105R2 

Ten Simple Rules for making a vocabulary FAIR

Dear Dr Cox,

I am pleased to inform you that your manuscript has been formally accepted for publication in PLOS Computational Biology. Your manuscript is now with our production department and you will be notified of the publication date in due course.

With kind regards,

Katalin Szabo
